# Physical Characteristics of and Transient Response from Thin Cylindrical Piezoelectric Transducers Used in a Petroleum Logging Tool

**DOI:** 10.3390/mi10120804

**Published:** 2019-11-22

**Authors:** Lin Fa, Nan Tu, Hao Qu, Yingrui Wu, Ke Sun, Yandong Zhang, Meng Liang, Xiangrong Fang, Meishan Zhao

**Affiliations:** 1School of Electronic Engineering, Xi’an University of Posts and Telecommunications, Xi’an 710121, Shaanxi, China; tn0918@126.com (N.T.); Wu_yingrui@126.com (Y.W.); xayddxsk@163.com (K.S.); zhangyandong@xupt.edu.cn (Y.Z.); focr@xupt.edu.cn (M.L.); fangxr@xiyou.edu.cn (X.F.); 2Baoding Hongsheng Acoustic-electric Equipment Co., Ltd., Baoding 071000, Hebei, China; ydfhcl@163.com; 3James Franck Institute and Department of Chemistry, The University of Chicago, Chicago, IL 60637, USA; llcai@uchicago.edu

**Keywords:** petroleum acoustical-logging, piezoelectric cylindrical-shell transducer, center-frequency, experimental-measurement

## Abstract

We report on a transient response model of thin cylindrical piezoelectric transducers used in the petroleum logging tools, parallel to a recently established transient response model of thin spherical-shell transducers. Established on a series of parallel-connected equivalent-circuits, this model provides insightful information on the physical characteristics of the thin cylindrical piezoelectric transducers, i.e., the transient response, center-frequency, and directivity of the transducer. We have developed a measurement system corresponding to the new model to provide a state-of-the-art comparison between theory and experiment. We found that the measured results were in good agreement with those of theoretical calculations.

## 1. Introduction

Acoustical measurement is ubiquitous in industrial applications, scientific research, and daily life, e.g., mobile and internet communication [1,2], exploration of underground mineral resources (oil, gas, coal, metal ores, etc.) [3], measurement of the in situ stresses of underground rock formation [4], and the inspection of mechanical properties of concrete [5,6], as well as intravascular ultrasound [7], medical imaging [8], biometric recognition [9], implantable microdevices [10], rangefinders [11], nondestructive detection [12,13,14], experimental verification of acoustic lateral displacement [15], inspection of a specific polarization state of a wave propagating in layered isotropic/anisotropic media [16,17], wave energy devices [18], and more. One of the key factors toward achieving a high-quality acoustic measurement is a good understanding of the properties of the acoustic transducers, e.g., the type of the transducers, the material property, and the geometric structure of the transducers. 

A unique characteristic of piezoelectric materials is their electric–mechanical transduction ability, converting mechanical energy to electrical energy and vice versa. This property has been exploited extensively in the construction of acoustic transducers for industrial applications [19], e.g., in electrical engineering, biomedical engineering, and geophysical engineering, among others. Along with technological progress, the quality of piezoelectric transducers has also been improving dramatically, e.g., smaller geometric dimensions, reduced noise level [20], lowered power consumption [21], etc. The typical geometric structures of acoustic transducers in practical applications are cylindrical, schistose, spherical, among others. Thin cylindrical transducers are widely used in industry, e.g., in petroleum logging tools. The physical properties of cylindrical piezoelectric transducers are also widely studied, including radiation, electric–acoustic and acoustic–electric conversions, and more. The radiation of a cylinder transducer with harmonic vibration was reported by Bordoni et al. [22] and Williams et al [23]. Fenlon [24] reported calculations for the acoustic radiation field at the surface of a finite cylinder using the method of weighted residuals. Wu [25] described an application of a variational principle for acoustic radiation from a vibrating finite cylinder. The effect of the length and the radius of a cylinder on its radiation efficiency was discussed and reported by Wang and coworkers [26,27].

We are concerned here with the thin cylindrical piezoelectric transducers used in petroleum logging tools, either as an acoustic source or as a receiver. Conventionally, in studies of acoustic-logging, the simplified analytical-models have been used for measuring acoustic signal waveforms and in processing the measured acoustic signal. Many times, the Tsang wavelet has been used as an acoustic source in acoustic logging [28,29]. Oversimplified acoustic-source functions have been used in forwarding-model research of acoustic logging and/or in inversion analysis and processing of the measured acoustic-logging signal, e.g., Ricker wavelet [30], Gaussian impulse wavelet [31], etc. However, the mathematical expressions of the simplified models have not been able to provide the practical relationship between a driving-voltage signal, geometrical and physical properties of the transducer, and radiated acoustic signal-wavelet. 

The transient response of a transducer driven by a sinusoidal electrical signal was reported by Piqtuette [32,33]. However, the reported results were not adequate for practical considerations for either acoustic logging or other acoustic measurements due to the driving-voltage signal of an exciting transducer containing multiple frequency components with different amplitudes and phases. In the process of providing a radiating acoustic signal, the transducer is also counteracted by the acoustic field radiated by itself and creates a radiation resistance and radiation mass, which are functions of the vibration frequency on the transducer’s surface. 

The complex effect of radiation resistance and radiation mass on the radiated acoustic-signal wavelet was reported by Fa and co-workers, where the deriving-voltage signal contained multi-frequency components. These researchers reported the case of thin spherical-shell transducers polarized in the radial direction, which radiated acoustic waves omnidirectionally [34,35,36,37]. Even though these reports are meaningful for modeling, the adopted transducers in the acoustic-logging tools and many other practical applications are mostly cylindrical, with radiation directivities quite different from that of the thin spherical-shell transducers. 

It is understood that correct analysis and inversion interpretation of the measured acoustic-logging signal wavelet rely on accurate acoustic measurement instruments, which must be established on a solid physical foundation with a strict engineering mechanism. Based on our current knowledge of the thin cylindrical piezoelectric transducers used in petroleum logging tools, further development is highly desirable to achieve an enhanced understanding to develop applicable measurement instrumentations. 

In this paper, we report a study of thin cylindrical transducers widely used in petroleum acoustic-logging tools. By adopting the method describing a thin spherical-shell transducer’s transient response [34] for the excited driving-voltage signal wavelet with multi-frequency components, we established the parallel-connected equivalent circuits for the thin cylindrical-shell transducers polarized in the radial direction. By solving the corresponding equations of motion, we analyzed the physical properties related to the transient response. Technically, we employed a measurement system with a high-resolution, high-sampling-rate digitizer to perform an experimental measurement, e.g., with a resolution range from 16-bit to 24-bit and a sampling rate from 500 KS/s to 15,000 KS/s. From this system, we were able to achieve good measurements in a large frequency range, even for weak acoustic signals. Based a proper analysis using the experimentally acquired data, we obtained the relationships between the various physical quantities, e.g., the driving electric-signal wavelet, the electric–acoustic/acoustic–electric conversion factors, the propagation media, and the measured acoustic signal. This, in turn, provided us with the ability to inspect the properties of the transducers, obtain the physical parameters of the measured fluid and solid material, check the quality of the measured objects, and perform a verification of existing and/or on-going scientific research. We report that the calculated results of the physical properties and transient response of the thin cylindrical-shell transducer were in good agreement with those of experimental observations.

## 2. Theory and Modeling

Let us consider a piezoelectric, thin, cylindrical transducer, with an average radius $\rho_{0}$ and a wall thickness $l_{t}$, polarized in the radial direction. The electrodes were connected to the inner and outer surfaces, as shown in Figure 1. Because the radius of the thin cylinder is much larger than its thickness ($\rho_{0}\gg l_{t}$), we have the following approximations: $\rho_{0}\approx \rho_{a}\approx \rho_{b}$ and $\rho_{0}=(\rho_{a}+\rho_{b})/2$ From the axis symmetry of the particle displacement, tangential and axial stresses are equal zero, i.e., $T_{\rho z}=T_{\rho \phi }=T_{z\phi }=0$.

If the inner and outer surfaces are free from any other forces, i.e., the normal stress in the radial direction is $T_{\rho }$ = 0, then the equation of motion for the particle vibrations of the transducer can be simplified to:(1)$$\rho_{n}\frac{\partial^{2}u_{\rho }}{\partial t^{2}}=-\frac{T_{\varphi }}{\rho_{0}},$$
(2)$$\rho_{n}\frac{\partial^{2}u_{z}}{\partial t^{2}}=-\frac{\partial T_{z}}{\partial z},$$where $T_{\phi }$ and $T_{z}$ are the normal stress in tangential and axial directions, respectively; *u_ρ_* and *u_z_* are the particle displacement in the radial and axial directions, respectively; and $\rho_{n}$ is the density of the transducer material.

Employing subscripts {1, 2, 3} in place of subscripts {*φ*, *z*, *ρ*}, the piezoelectric equations with respect to radial polarization can be expressed as:(3)$$S_{1}=s_{11}^{E}T_{1}+d_{31}E_{3},$$
(4)$$D_{3}=d_{31}T_{1}+\varepsilon_{33}^{T}E_{3},$$where $S_{1}$ and $T_{1}$ are the strain and stress in *φ*-direction, respectively; $D_{3}$ and $E_{3}$ are the radial components of the electric displacement and electric field vectors, respectively; and $s_{11}^{E}$, $\varepsilon_{33}^{T}$, and $d_{31}$ are the compliance, piezoelectric, and dielectric constants of the piezoelectric material, respectively. Because the height (*H*) of the transducer is much larger than its thickness, the coupling between the axial and radial vibrations can be neglected and the axial stress $T_{2}$ can be ignored. The vibration of the thin cylindrical transducer can be simplified as being one-dimensional in the radial-direction. 

Substituting Equation (3) into Equation (1) yields:(5)$$m\frac{d^{2}u_{\rho }}{dt^{2}}=-\frac{2\pi Hl_{t}}{s_{11}^{E}}S_{1}+\frac{2\pi Hl_{t}d_{31}}{s_{11}^{E}}E_{3},$$where $m=2\pi Hl_{t}\rho_{0}\rho_{n}$ is the mass of the transducer.

We set the transducer to be in coupling fluid, where only its outer surface was in contact with the coupling fluid. The transducer vibrates in the radial direction, where the vibration of the thin cylindrical transducer’s outer surface causes the surrounding medium to expand and contract alternately in the radial direction. Consequently, the acoustic waves are radiated outward. Meanwhile, the thin cylindrical transducer is in the acoustic field radiated by itself. Then, it is acted on by a counterforce caused by the acoustic field. On the outer surface, by using a similar method described in References [34,35], we can obtain this counter-force as follows:(6)$$F_{r}=-R\left(\frac{k^{2}\rho_{0}^{2}}{1+k^{2}\rho_{0}^{2}}+j\frac{k\rho_{0}}{1+k^{2}\rho_{0}^{2}}\right)\frac{du_{\rho }}{dt}=-\left(R_{\rho }+iX_{\rho }\right)\frac{du_{\rho }}{dt}=-Z_{\rho }\frac{du_{\rho }}{dt}\;R_{\rho }=R\frac{k^{2}\rho_{0}^{2}}{1+k^{2}\rho_{0}^{2}},\;X_{\rho }=R\frac{k\rho_{0}}{1+k^{2}\rho_{0}^{2}},$$where $R=2\pi \rho_{0}H\rho_{m}v_{m}$; $\rho_{m}$ and $v_{m}$ are density and acoustic velocity of coupling fluid around the transducer, respectively; *i* is the unit imaginary number; and the radiation resistance and the radiation reactance are $R_{\rho }$.

If the thin cylindrical transducer vibrates harmonically with various frequencies, its radiation resistance and radiation reactance would also be different, where $k=\omega /v_{c}$.

Due to the viscosity of the coupling liquid, the vibration of the transducer creates a frictional resistance force, which can be expressed as:(7)$$F_{f}==-R_{m}\frac{du_{\rho }}{dt},$$where $R_{m}$ is the frictional resistance on the surface of the transducer, which is proportional to the viscosity coefficient of the coupling fluid and the outer side wall area of the transducer. The transducer’s radiation surface is approximately $A\approx 2\pi \rho_{b}H$, and the total outer force acted on the transducer is:(8)$$F=F_{r}+F_{f}=-(R_{\rho }+R_{m}+iX_{\rho })\frac{du_{\rho }}{dt}.$$

The axial symmetry of thin-cylindrical transducer leads to:(9)$$S_{1}=\frac{u_{\rho }}{\rho_{0}}.$$

For the harmonic vibration, substituting Equations (8) and (9) into Equation (5) yields: (10)$$u_{r}=\frac{2\pi Hl_{t}d_{31}/s_{11}^{E}}{-\omega^{2}\left(m+m_{\rho }\right)+j\omega \left(R_{\rho }+R_{m}\right)+1/C_{m}}E_{3},$$where $C_{m}=\rho_{0}s_{11}^{E}/(2\pi Hl_{t})$.

Also, from Equations (3) and (4), we have:(11)$$D_{3}=\frac{d_{31}}{s_{11}^{E}\rho_{0}}u_{\rho }+\varepsilon_{33}^{T}(1-K_{31}^{2})E_{3}$$where $K_{31}=d_{31}^{}/\sqrt{s_{11}^{E}\varepsilon_{33}^{T}}$.

Because the thickness of the cylindrical transducer is small enough, the edge effect of the upper and lower cross-section can be neglected. From the Gauss theorem, the total charge on each electrode (either inner or outer surface) of the thin cylindrical transducer is $Q=2\pi \rho_{0}HD_{3}$. The instantaneous current into the electrodes is the time derivative of *Q*. For a harmonic driving electric signal, we have:(12)$$I=\frac{dQ}{dt}=i\omega C_{0}V+N^{2}\frac{V}{R_{\rho }+R_{m}+i\omega (m+m_{\rho })+1/i\omega C_{m}},$$where $C_{0}=2\pi \rho_{0}H\varepsilon_{33}^{T}(1-K_{p}^{2})/l_{t}$; $N=2\pi Hd_{31}/s_{11}^{E}$, which is the electric–mechanical turn coefficient of the thin cylindrical transducer; and $V=l_{t}E_{r}$, which is the voltage across the two electrodes of the thin cylindrical transducer.

Suppose that the driving circuit outputs a sinusoidal signal $U_{1}(t)$ with an angular frequency ω and output resistance *R_o_*. Based on Equation (12), the electric–acoustic equivalent circuit of the transducer for harmonic vibration is shown in Figure 2a. Its corresponding *s*-domain network is shown in Figure 2b, where $v_{\rho }(t)$ and $v_{\rho }(s)$ are defined as particle vibration speeds at the transducer’s surfaces, and $m_{\rho }$ is defined as the transducer’s radiation mass.

At the electric terminals of the *s*-domain in the network, as shown in Figure 2b, we have: (13)$$U_{1}(s)=V(s)+I(s)R_{0}$$and(14)$$I(s)=sC_{0}V(s)+I_{1}(s)=sC_{0}V(s)+Nv_{r}(s).$$

The transient response process of the thin cylindrical transducer can be held as a zero-state response. In the *s*-domain, we define the electric–acoustic conversion system function as a ratio of the vibration speed of the transducer’s outer surface to the sinusoidal driving voltage signal. In terms of Figure 2b and Equations (13) and (14), this electric–acoustic conversion function can be expressed as:(15)$$H_{1}(s)=\frac{v_{r}(s)}{U(s)}=\frac{ds}{s^{3}+as^{2}+bs+c},$$where:$$a=\frac{R_{m}+R_{\rho }}{m+m_{\rho }}+\frac{1}{R_{0}C_{0}},\;b=\frac{R_{m}+R_{\rho }}{(m+m_{\rho })R_{0}C_{0}}+\frac{1}{(m+m_{\rho })C_{m}}+\frac{N^{2}}{(m+m_{\rho })C_{0}},\;c=\frac{1}{(m+m_{\rho })C_{0}C_{m}R_{0}},\;d=\frac{N}{(m+m_{\rho })C_{0}R_{0}}.$$

By applying the residue theorem to Equation (15), we obtain the electric–acoustic impulse response of the thin cylindrical transducer:(16)h1(t)=∑i=1LRes[H1(si)esit],where *L* is the pole number of Equation (15). The denominator of this equation is a cubic polynomial with one unknown variable with the three roots:(17)$$s_{1}=x+y-a/3,$$
(18)$$s_{2,3}=-(x+y)/2-a/3\pm j\sqrt{3}(x-y)/2,$$where: $$x=\sqrt[{3}]{-q/2+\sqrt{D}},\;y=\sqrt[{3}]{-q/2-\sqrt{D}},\;p=b-a^{2}/3,\;q=c+2a^{3}/27-ab/3,\;D={\left(p/3\right)}^{3}+{\left(q/2\right)}^{2}.$$

In theory, there are three cases with different D parameters—*D* > 0, *D* = 0, and *D* < 0—which correspond to the three motion modes: over-damping, critical-damping, and under-damping (oscillatory), respectively. Practically, the physical properties of a piezoelectric material, i.e. its physical and piezoelectric parameters guarantee that the parameter *D* is greater than zero, means that the transducer works only in the oscillatory mode and its electric–acoustic impulse response can be written as:(19)$$h_{1}(t)=A_{3}\exp\left(-\alpha_{1}t\right)+B_{3}\exp\left(-\beta_{1}t\right)\cos\left(\omega_{1}t+\varphi_{1}\right)$$where: $$A=(x+y)/2,\;B=(x-y)/2,\;\beta_{1}=A+a/3,\;\alpha_{1}=a/3-2A,\;\sigma_{1}=\beta_{1}-\alpha_{1},\;A_{3}=\frac{-d\alpha_{1}}{\sigma_{1}^{2}+3B^{2}},\;B_{3}=-\frac{d(\sigma_{1}-\beta_{1})}{\sigma_{1}^{2}+3B^{2}},\;\omega_{1}=\sqrt{3}B,\;\varphi_{1}=\arctan\frac{\beta \sigma_{1}+3B^{2}}{\sqrt{3}B(\sigma_{1}-\beta_{1})}.$$

The acoustic–electric conversion of the transducer is the inverse process of the electric–acoustic conversion. By repeating the discussed process in reverse order, we obtain the acoustic–electric impulse response of the transducer as follows:(20)$$h_{3}(t)={\bar{A}}_{3}\exp[-\alpha_{3}t]+{\bar{D}}_{3}\exp[-\beta_{3}t]\cos(\omega_{3}t+\varphi_{3}),$$where:$$\bar{a}=\frac{\left(m+m_{r})+C_{o}R_{o}(R_{r}+R_{m}\right)}{C_{o}R_{o}(m+m_{r})},\;\bar{b}=\frac{C_{o}R_{o}+C_{m}(R_{r}+R_{m})+C_{m}R_{o}N^{2}}{C_{m}C_{o}R_{o}(m+m_{r})},\;\bar{c}=\frac{1}{C_{m}C_{o}R_{o}(m+m_{r})},\;\bar{d}=\frac{\rho_{0}v_{c}N}{C_{o}(m+m_{r})},\;\bar{x}=\sqrt[{3}]{-\bar{q}/2+\sqrt{\bar{D}}},\;\bar{y}=\sqrt[{3}]{-\bar{q}/2-\sqrt{\bar{D}}},\;\bar{p}=\bar{b}-{\bar{a}}^{2}/3,\;\bar{q}=\bar{c}+2{\bar{a}}^{3}/27-\bar{a}\bar{b}/3,\;\bar{D}={\left(\bar{p}/3\right)}^{3}+{\left(\bar{q}/2\right)}^{2},\;\bar{A}=(\bar{x}+\bar{y})/2,\;\bar{B}=(\bar{x}-\bar{y})/2,\;\beta_{3}=\bar{A}+\bar{a}/3,\;\alpha_{3}=\bar{a}/3-2\bar{A},\;\sigma_{3}=\beta_{3}-\alpha_{3},\;{\bar{A}}_{3}=\frac{-\bar{d_{3}}\alpha_{3}}{\sigma_{3}^{2}+3{\bar{B}}^{2}},\;{\bar{B}}_{3}=-\frac{\bar{d_{3}}(\sigma_{3}-\beta_{3})}{\sigma_{3}^{2}+3{\bar{B}}^{2}},\;{\bar{D}}_{3}=\sqrt{{\bar{B}}_{3}^{2}+{\bar{C}}_{3}^{2}},\;{\bar{C}}_{3}=\frac{-\bar{d_{3}}(\beta_{3}\sigma_{3}+3{\bar{B}}^{2})}{\sqrt{3}\bar{B}(\sigma_{3}^{2}+3{\bar{B}}^{2})},\;\omega_{3}=\sqrt{3}\bar{B},\;\varphi_{3}=\arctan\frac{\beta_{3}\sigma_{3}+3{\bar{B}}^{2}}{\sqrt{3}\bar{B}(\sigma_{3}^{}-\beta_{3})}.$$

For a harmonic vibration, in the equivalent circuits in Figure 2, the two mechanical components, i.e., radiation resistance and radiation mass, are functions of vibration frequency. Also, for most cases, either the electrical signal of an exciting source transducer or the acoustic signal arriving at the receiver transducer is a signal wavelet with multi-frequency components. Excited by an electric/acoustic signal-wavelet with multi-frequency components, the vibration of the transducer’s surface also consists of multiple sinusoidal frequency components. 

The Fourier transform of the electric/acoustic signal of an excited transducer can be expressed as a linear superposition of sine-wave components with different frequencies, amplitudes, and phases. The electric–acoustic/acoustic–electric excitation can be processed by the parallel-connected network as shown by parts I and III in Figure 3. Each of these equivalent circuits in the network has its own unique electric–acoustic/acoustic–electric impulse response resulting from its individual radiation resistance and radiation mass. 

A continuous driving electric signal $U_{1}(t)$ with amplitude spectrum S(ω) and phase spectrum ϕ(ω) can be decomposed into *N* frequency components using an *N*-point discrete Fourier transform. Each frequency component can be written as: (21)$$U_{1j}(t)=\left|S(\omega_{j})\right|\cos\left[\omega_{j}t+\phi (\omega_{j})\right]$$where *j* = 1, 2, 3, …, *N*; and $\left|S(\omega_{j})\right|$ and $\phi (\omega_{j})$ are the amplitude and the phase of the *j*th sinusoidal frequency component. Therefore, a normalized driving electric signal can be expressed as:(22)$$U_{1}(t)=\sum_{j=1}^{N}U_{1j}(t)/\max\left[\sum_{j=1}^{N}U_{1j}(t)\right].$$

The output from the *j*th circuit in part I of Figure 3 is a convolution of the *j*th sinusoidal frequency component of the driving electric signal, with the *j*th electric–acoustic impulse response function being:(23)$${v_{r1j}(t)|}_{\omega_{j}}={\left[U_{1j}(t)*h_{1j}(t)\right]|}_{\omega_{j}}.$$

Then, the normalized vibration speed of the surface of the thin cylindrical transducer, i.e., the radiated acoustic signal, is defined as: (24)$$v_{r1}(t)=\sum_{j=1}^{N}{v_{r1j}(t)|}_{\omega_{j}}/\max[\sum_{j=1}^{N}{v_{r1j}(t)|}_{\omega_{j}}].$$

For the *j*th frequency component of the radiated acoustic signal, if the propagation medium produces an acoustic impulse response $h_{2j}(t){|}_{\omega_{j}}$, it would yield the *j*th frequency component arriving at the receiver transducer as:(25)$$v_{r3j}(t,\omega_{j})={\left[v_{r1j}(t)*h_{2j}(t)\right]|}_{\omega_{{}_{j}},t_{{}_{1j}}},$$where $t_{1j}$ is the propagation time of the *j*th sinusoidal frequency component from the source transducer to the receiver transducer.

The acoustic–electric conversion of a transducer is the inverse of the electric–acoustic conversion. The *j*th frequency component of an acoustic signal arriving at the receiver transducer passing the *j*th circuit (part III of Figure 3) is converted to an electric signal according to: (26)$$U_{3j}(t)={\left[v_{r3j}(t)*h_{3j}(t)\right]|}_{\omega_{j},t_{1j}}.$$

Finally, the measured acoustic signal, i.e., the electric signal at the electric terminals of the receiver transducer, is a collection of the outputs from all circuits in part III of the network (Figure 3), which is normalized as:(27)$$U_{3}(t)=\sum_{j=1}^{N}U_{3j}(t){|}_{\omega_{j}}/\max[\sum_{j=1}^{N}U_{3j}(t){|}_{\omega_{j}}].$$

The above discussion shows that an acoustic-measurement process can be achieved through a parallel-connected transmission network, as shown in Figure 3.

## 3. Calculation

We used a piezoelectric material PZT-5H (Lead Zirconate Titanate-5H, Baoding Hongsheng Acoustic-electric Equipment Co., Ltd., Baoding, Hebei 071000, China) in the construction of the cylindrical transducers. The physical parameters of the piezoelectric material PZT-5H are $\varepsilon_{33}^{T}=300.9\times 10^{-10}$ F/m^2^, $s_{11}^{E}=16.5\times 10^{-12}$ m^2^/N, $d_{31}=-274\times 10^{-12}\;m/V$, and $\rho_{n}=7.5\times 10^{3}$ kg/m^3^. silicone oil was used as the coupling medium around the transducers, where the physical parameters are $v_{m}$ = 1424 m/s and $\rho_{m}=856$ kg/m^3^ (Shandong Longhui Chemical Co., Ltd, Jinan, Shaandong 250131, China).

### 3.1. Relationship between the Center Frequency versus the Radius for a Thin Cylindrical Transducer

As described above, the transducer always works in an oscillatory mode. As a reference, we defined a free-mechanical-load transducer as a transducer in a vacuum, i.e. $R_{\rho }$ = 0, $m_{\rho }$ = 0, and $R_{m}$ = 0. The center frequency of the piezoelectric PZT-5H thin cylindrical transducer as a function of average radius is presented in Figure 4, along with a case of the mechanical load, i.e., the transducer in the transformer oil, where $R_{\rho }$ ≠ 0, $m_{\rho }$ ≠ 0, and $R_{m}$ ≠ 0.

Figure 4 shows that: (i) the transducer’s center frequency decreased with respect to its increased radius, with or without a mechanical load; (ii) the center frequency of the mechanical load was lower than that of the free mechanical load; and (iii) the mechanical load effect on the transducer’s center frequency decreased with the increased radius.

### 3.2. Relationship between the Center Frequency versus a Forced Vibrational Frequency

We built two piezoelectric PZT-5H shin-cylindrical transducers polarized in the radial direction, with an average radius $\rho_{0}$ = 20.5 mm, height *H* = 6.0 mm, and wall thickness $l_{t}=\rho_{0}/8$. Figure 4 shows that the center frequency of the transducers with a mechanical load was 20.5180 kHz. We also determined that the center frequency of the free-mechanical-load transducer was at 22.5120 kHz. 

For the case of the mechanical load, we selected the parameter $R_{m}=$ 0.2*R*. When normalized by the transducer’s free-load center frequency, for the transducer’s forced harmonic vibrational motions at several frequencies, the electric–acoustic/acoustic–electric conversion properties were calculated and are presented in Figure 5.

At the center frequency of 22.5120 kHz for the free mechanical load, both the source and receiver transducers had a maximum transition amplitude, which was the largest for all cases. For the mechanical loaded transducers with harmonic forced vibrations, a lower forced vibration frequency corresponded to a lower center frequency with a larger maximum transition amplitude. With the increased vibration frequency going close to the free-mechanical-load resonance frequency $f_{0}$, a mechanical-loaded transducer showed a maximum transition amplitude, but it was much smaller than that of the thin cylindrical transducer free vibration. 

The observations above confirmed our understanding that the electric–acoustic/acoustic–electric conversion of the transducer was dependent not only on the physical and geometrical parameters of a transducer and the physical parameters of the medium around the transducer, but also the forced harmonic vibration frequency. The radiation resistance and radiation mass were parameters affecting the forced vibration frequency, which led to the variations of the electric–acoustic/acoustic–electric conversion properties of the transducer.

### 3.3. Radiation Directivity of a Thin Cylindrical Transducer

Different from the thin-shell spherical transducer, the radiation property of a thin cylindrical transducer has a determinative directivity, which can be described using [8]:(28)$$G(\theta )={\left[\frac{J_{0}^{2}(\frac{2\pi \rho_{a}}{\lambda }\cos\theta )+\cos^{2}\theta J_{1}^{2}(\frac{2\pi \rho_{a}}{\lambda }\cos\theta )}{J_{0}^{2}(\frac{2\pi \rho_{a}}{\lambda })+J_{1}^{2}(\frac{2\pi \rho_{a}}{\lambda })}\right]}^{1/2}\frac{\sin(\frac{\pi H}{\lambda }\sin\theta )}{\frac{\pi H}{\lambda }\sin\theta },$$where $J_{0}$ is the zeroth-order Bessel function; $J_{1}$ is the first-order Bessel function; λ is the wavelength of the acoustic wave in coupling fluid, i.e., transformer oil; and θ is the angle between the propagation direction of radiated acoustic signal and the normal direction of the thin cylindrical transducer. 

From Equation (28), the calculated radiation directivity of a thin cylindrical transducer is shown in Figure 6 It is shown that the acoustic energy radiated by the transducer was more centralized in the normal direction with the increased height of the thin cylindrical transducer. 

The effective radiating area of a cylindrical transducer, in the direction of *θ*, may be defined as the product of the area of the thin cylindrical transducer $S_{a}(=2\pi \rho_{a}H)$ and the directivity *G*(*θ*):(29)$$ERA=S_{a}\cdot G(\theta )=2\pi \rho_{a}H\cdot G(\theta ).$$

For a given cylindrical transducer with a fixed average radius, a normalized effective radiating area is labelled as *NERA*. The relationship of the normalized *NERA* versus *H* in several different directions are presented in Figure 7. The calculated results indicated that the transducer’s effective radiating area did not increase monotonously with respect to its height or its radiation area. This was exactly the effect of the transducer’s radiation directivity, which was a function of the radiation direction. With this understanding, by selecting a suitable height of the transducer, we may be able to achieve an increased effective radiation area of the cylindrical transducer for a set special direction.

### 3.4. Transient Response of a Cylindrical Transducer Excited by a Signal with Multi-Frequency Components

As an example, let us consider a gated sinusoidal electric signal as the source of excitation with multi-frequency components as:(30)$$U_{1}(t)=\left[H(t)-H(t-t_{0})\right]U_{0}\sin\left(\omega_{s}t\right),$$where, $U_{0}$, $\omega_{s}$, and $t_{0}$ are the amplitude, angular frequency, and time window of the driving electric voltage signal, respectively; and H(⋅) is the Heaviside unit step function. A simple Fourier transform yields the source signal in the frequency domain as: (31)$$S_{1}(\omega )=U_{0}\{\omega_{s}-(\omega_{s}\cos\omega_{s}t_{0}+j\omega \sin\omega_{s}t_{0})\exp[-j\omega t_{0}]/(\omega_{s}^{2}-\omega^{2})$$with a corresponding phase spectrum: (32)ϕ(ω)=tg−1[Im(S1)Re(S1)].

Now, we selected the source signal parameters $U_{0}=1$ V, $t_{0}=4/f_{s}$, and $f_{s}=\omega_{s}/2\pi =$ 20.5180 kHz, which was the mechanical-load center frequency of the transducer. The center frequency of this driving electric signal was then 20.1690 kHz, which was slightly smaller than the value of $f_{s}$.

Figure 8 shows that the theoretical waveform of the driving electric voltage signal agreed well with the waveform synthesized using a discrete Fourier transform. In turn, it guaranteed the accuracy of our analysis in the following sections.

The gated sinusoidal driving electric signal was expanded on the basis of a series of sine-waves with different frequencies, amplitudes, and phases. Each of these sinusoidal components, as an individual excitation source, was applied to the parallel circuits of Figure 3. The output signal from each parallel circuit in the network (see Figure 3) was calculated and analyzed. 

The cumulative output of the waveform of the parallel circuits in part I of Figure 3 is presented in Figure 9a. The corresponding amplitude spectrum is presented in Figure 9b, which shows that the center frequency of the radiated acoustic signal was 20.2500 kHz. It was smaller than the load center frequency 20.5180 kHz, but larger than that of gated sinusoidal driving electric signal at 20.1690 kHz, which was the result of the joint action of the electric–acoustic conversion through the source transducer and the electric driving signal.

The studied source transducer was cylindrical and had a radiation directivity. If we assumed that water is an ideal elastic medium, then the acoustic signal propagating inside the water could have a geometrical attenuation but not a viscous attenuation. Additionally, all frequency components of the acoustic signal would propagate at the same speed. The shapes of the waveform and frequency spectrum would not change. The amplitude would decrease with respect to the increased propagation distance only. Under these conditions, we calculated the signals at the electric terminals of the receiver transducer. 

For a thin cylindrical transducer, the shape of its radiation directivity is the same as that of its receiving directivity; therefore, the acoustic impulse response in water can then be written as:(33)$$h_{2}(t)=G_{1}(\theta_{1})G_{2}(\theta_{2})\delta \left(t-t_{1}\right)/(1+r),$$where $h_{21}\left(t,\omega_{1}\right)=\ldots =h_{2j}\left(t,\omega_{j}\right)=\ldots h_{2N}\left(t,\omega_{N}\right)=h_{2}(t)$, *t*_1_ is the propagation time and *r* is the distance of the radiated signal in water, $\theta_{1}$ is the angle between of the propagation direction of the radiated acoustic wave and the normal direction of the side wall of the source-transducer, $G_{1}(\theta_{1})$ is the corresponding radiation directivity at this propagation direction ($\theta_{1}$), $\theta_{2}$ is the angle between the direction of the acoustic wave propagating to the receiver transducer and the normal direction of the side-wall of the receiver-transducer, and $G_{2}(\theta_{2})$ is the corresponding receiving directivity.

To determine the quality of a transducer for electric–acoustic/acoustic–electric conversion, the most important piece of information comes from the analysis of the output signal following electric–acoustic–electric transduction. To accomplish this operation, we set $\theta_{1}=\theta_{2}=0^{\circ }$ and the distance from the source transducer to the receiver transducer was 1.0 m. Applying the gated sinusoidal driving electric signal to the source transducer, we calculated the cumulative output signals on the receiver transducer, as shown as solid lines in Figure 10. 

Figure 10 shows the cumulative signals of the waveform and amplitude spectrum at the electric terminals of the receiver transducer. From the theoretical calculation, the center frequency of the output signal at the electrical terminals of the receiver transducer was 20.5000 kHz. This center frequency was slightly smaller than the transducer’s load center frequency and slightly greater than the center frequency of the acoustic signal radiated by the source transducer. We also believe that this was the result of the joint action of a lower center frequency of the radiated acoustic signal and a higher center frequency of the transducer with a mechanical load. 

For an average radius of 20.5 mm, Figure 4 shows that the calculated center frequency of the mechanical-loaded transducer was 20.5800 kHz and the measured center frequencies of the two cylindrical transducers with the same size were 19.6780 kHz and 20.7520 kHz. The waveform and amplitude spectrum of the measured acoustic signal, i.e., the measured electric signal at the electric terminals of the receiver transducer, are shown as the dotted lines in Figure 10. Overall, the theoretical calculation had a good agreement with the experimental measurement. 

### 3.5. Influence of the Transducer’s Radiating/Receiving Directivity on the Measured Acoustic Signal 

Now, let us consider a measurement system, where a cylindrical transducer is used as an acoustic source and four cylindrical transducers as a receiver array, as shown in Figure 11. The transition media of this system is water. The waveforms of the signals at the electric terminals of the four cylindrical transducers in the receiver array were calculated and are presented in Figure 12. 

The results of the calculation show that the greater the deflection angle, the greater the influence on the signal at the electric terminals of each cylindrical transducer in the receiver array. This was especially true in the near-field area. The amplitude of the signal at the electric terminals decreased with increased angle $\theta_{j}$.

## 4. Devices and Measurements 

To rationalize the proposed transient response and physical properties of the cylindrical transducers, we developed an experimental measurement system (see Appendix A). In particular, this system consisted of a mechanical assembly, an electrical hardware module, and a system software module aimed at system control, measurement, and analysis. A structure flowchart is presented in Figure 13. 

### 4.1. Mechanical and Electrical Hardware 

The mechanical parts include steering engines, stepping motors, and sliding rails. The combined assembly with a microcontroller formed a positioning platform, which was used to slide and/or rotate the source/receiver transducers to the proper positions and directions in a silencing tank filled with water. The silencing tank was used to gauge the physical properties of the transducer, e.g., the electric–acoustic/acoustic–electric transient response, directivity, radiation power, and receiving sensitivity of the transducer. 

The electric hardware was composed of a microcontroller, an electric-signal waveform generator, a power amplifier, a high-resolution and high-sampling-rate digitizer (up to 24 bit and 15 MHz), and a desktop/laptop for central system control.

### 4.2. System Software

The system software was developed on the LabVIEW platform in Graphic Programming Language (G-code). Communications between the computer and the hardware were realized through USB serial ports. Powered by a graphic interface control panel, users have the options of selecting different driving signals, configuring the power amplifier, adjusting the position and rotational angle of the transducers, and attaining data acquisition of the measured acoustic signal. 

The developed software consists of four functional modules, as shown in Figure 14. On a graphic computer interface panel, the VISA (Virtual Instrument Software Architecture) library from the LabVIEW platform was employed to achieve the control of the four modules through serial port communications. In the function/selection boxes in the human–computer interface modules, users input the parameters and commands for special operations and acoustic measurements.

#### 4.2.1. Electric Signal Module 

In controlling the widgets, knobs, and switches on the software interface panel, this module configures and modulates the types of signal and frequency, amplitude, and the cycle of an electric source signal. The options for the electric source signal are the gated sinusoidal wave, square wave, triangular wave, and sawtooth wave. 

#### 4.2.2. Power Amplification Module

This module configures the amplification gain of a power amplifier, which amplifies the electric signal created by the electric signal waveform generator. The amplified electric signal is used to excite the source transducer to radiate the acoustic signal. This is achieved through a rotary button on the interface control panel. For an output voltage signal with a frequency belt from 0.15 MHz to 1.5 MHz, the maximum peak-to-peak value can reach 220 V. Meanwhile, for an output voltage signal with a frequency range from 10 kHz to 150 kHz, the maximum peak-to-peak value can be up to 1600 V. For an electric signal source with general impedance, a suitable input impedance is 50 Ω. For an electric signal source with high impedance, an appropriate input impedance is 50 kΩ.

#### 4.2.3. Display/Storage Module

Using this module, users may configure the acquisition channel, sampling rate, and amplitude range of an acquired signal. Data in the time–frequency domain acquired by the digitizer are sent to the interface panel through a USB port. These data can be displayed on screen in real-time or stored for later analysis. The format of the acquired data can be in either simple text or in a binary form.

#### 4.2.4. Motion/Rotation Module

This module automatically initializes the position and direction of the source/receiver transducers, which means the sliding unite moves to the zero position and the steering engines return to the direction of zero degrees. Users can configure the stepping distance of the sliding unit and the rotation angle of the steering engine by entering specific parameters on the interface control panel. The available stepping distance is 0.0 to 3000.0 mm and the range of the rotating angle of the steering engine is 0 to 360 degrees.

### 4.3. System Workflow

Through the control panel with a graphic interface, users can manage the operation of the mechanical and electric hardware; complete data acquisition; and perform calculations, analysis, display, and storage of the measured data. 

The first step is to perform a system alignment through the control panel operation. Based on the specific need of a given measurement, users may send a specific command to the microcontroller to initialize the system. The stepping motors and steering engines are adjusted on the sliding trails to achieve a specific setting in the vertical and horizontal directions, as well as a set angle relative to the horizontal plane. This operation will make the source/receiver transducers move to a desired position and rotate to a set direction for the acoustic measurement. 

Through the control panel, users may send a command to the electric-signal generator to send an electric source signal to the power amplifier, e.g., a gated sinusoidal wave, a square wave, etc. The amplified electric signal excites the source transducer to emit an acoustic signal outward. The acoustic signal then propagates to the receiver through a specific medium or a measured object. Finally, by going through the receiver transducer, the acoustic signal can be converted to an electric signal. The final output is acquired by a high-resolution, high-sampling-rate digitizer and sent back to the computer for display, calculation, analysis, and storage.

The two specially fabricated piezoelectric PZT-5H thin-cylindrical transducers were set to be polarized in the radial direction. They had the same geometrical structure, size, and inner radius. The radius, thickness, and height of the transducers were 20.5 mm, 3 mm, and 50 mm, respectively.

The center frequencies of the thin cylindrical transducers measured using an impedance analyzer PV90A (Bander Electronics Co., Ltd., Beijing, 244021, China) were 20.7520 kHz and 19.6780 kHz with a mechanical load, as shown in Figure 4, as noted by the two five-pointed stars, while the calculated center frequency of the source transducer with a mechanical load was at 20.5180 kHz (see Figure 5). The center frequencies of these two cylindrical transducers were slightly different and close to the center frequency found from the theoretical calculations. We contributed the center frequency difference between the two cylindrical transducers to manufacturing imperfection. 

We assembled the cylindrical transducers in a silencing tank filled with water, one as a source transducer and the other as a receiver transducer. The distance between the transducers was set to be 1.0 m. A gated sinusoidal electric signal was generated by a command from the control panel to the waveform generator. Powered by an amplifier, the sinusoidal electric signal provided excitation to the source transducer to emit an acoustic signal. The measured acoustic signal in the time and frequency domains are shown as the dotted lines in Figure 10, i.e., the electric signal at the electric terminals of the receiver transducer. The center frequency of the signal at the electrical terminals of the receiver transducer was analyzed and compared between the theoretical calculation and experimental measurement, which yielded 20.5000 kHz and 19.4440 kHz, respectively. These values are somewhat different from those of the source transducer. This small discrepancy was likely the result of the acoustic–electric filtering effect on acoustic signal arriving at the receiver transducer.

As shown in Figure 10, the shapes of the waveform and the amplitude spectrum of the measured acoustic signal had a noticeable discrepancy from those of the theoretical calculations, even though there was good qualitative agreements in form and shape. Another likely culprit was that only half of the surrounding area of the transducer was receiving the acoustic signal from the source transducer in a nonuniform manner (see Figure 13). 

## 5. Final Remarks

Through modeling, calculation, and experimental measurements, we achieved an enhanced understanding of the transient responses of the electric–acoustic and acoustic–electric conversions based on the piezoelectric cylindrical transducers, which are widely used in petroleum acoustic logging. 

For cylindrical transducers, the process of electric–acoustic conversion is the reciprocal process of acoustic–electric conversion. In the processes of these conversions, there are three possible mathematical solutions corresponding to three different physical states, i.e., overdamping, critical-damping, and the oscillatory mode. Practically, the oscillatory mode is the only physically meaningful state for the piezoelectric cylindrical transducers.

For signal conversions, the driving signals applied to stimulate the transducers contained multiple frequency components. This was the reality for both the driving electric signal to the transducer’s electric terminals and the acoustic signal to its mechanical terminals. Meanwhile, each frequency component was processed independently and the overall signal transmission was achieved as a superposition of all the individual processes. To achieve this goal, we constructed an equivalent circuits network, where each circuit processed an individual frequency component. 

In constructing the circuits for the cylindrical transducers, we noted that radiation resistance and radiation mass were functions of the frequency. One of the simplest processes was to establish a set of equivalent circuits under the harmonic vibrational motions. The acoustic-measurement process could be viewed as a transmission system network with multiple parallel-connected equivalent circuits. 

To achieve an enhanced understanding of the electric–acoustic and acoustic–electric conversions, we need to explore many important conversion properties, e.g., the impulse response, amplitude spectrum, center frequency, and more. These properties are influenced not only by the physical and geometrical parameters of the transducer, and the physical parameters of the transition medium, but also by the property and type of the driving electric signal. Obviously, the measured acoustic signal in both the time and frequency domains are dependent on the electric–acoustic and acoustic–electric conversion property and the properties of the driving electric signal. 

The directivity of the thin cylindrical transducers can be significant for acoustic measurement. Unlike the spherical shell transducers, which are polarized in the radial direction, the directivity is a unique property of the thin cylindrical transducers, which provides a none-ignorable influence on the measured acoustic signal. It influences the optimal effective radiation area of the thin cylindrical transducers, which varies with the height of the cylindrical transducers.

In summary, for a transient response and relevant physical properties of the piezoelectric, thin cylindrical transducers used in petroleum acoustic logging tools, we established a new transient response model and developed a corresponding measurement system to perform an experimental check on the theoretical calculations. The newly developed experimental system had a high measurement resolution and a high sampling rate, which was able to achieve a considerable measurement precision. It should be able to be applied for both scientific research and for industrial applications.

## Figures and Tables

**Figure 1 micromachines-10-00804-f001:**
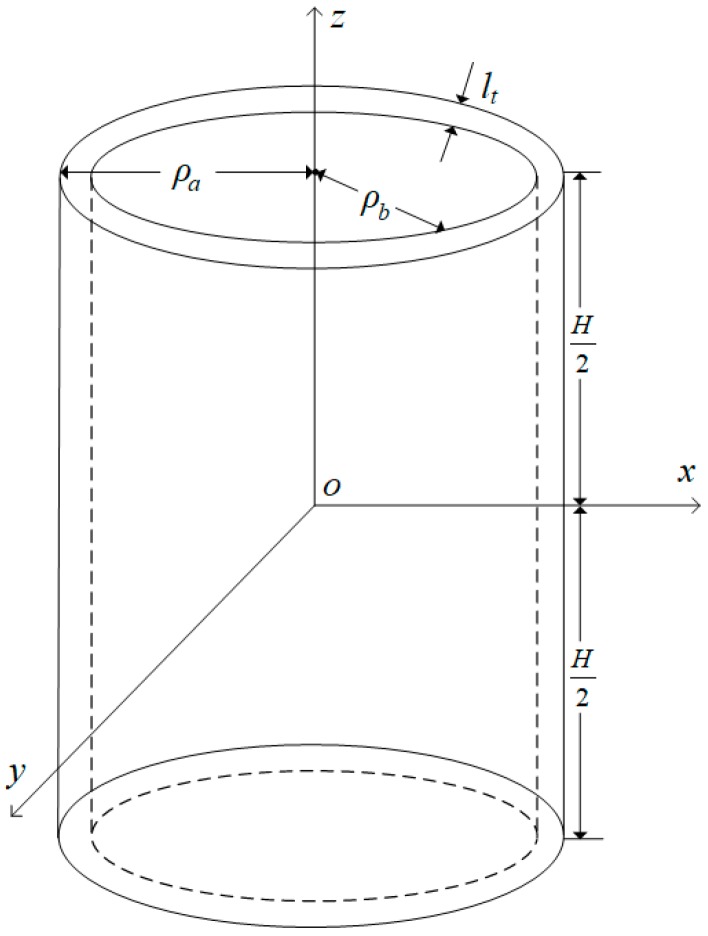
A piezoelectric thin spherical-shell transducer.

**Figure 2 micromachines-10-00804-f002:**
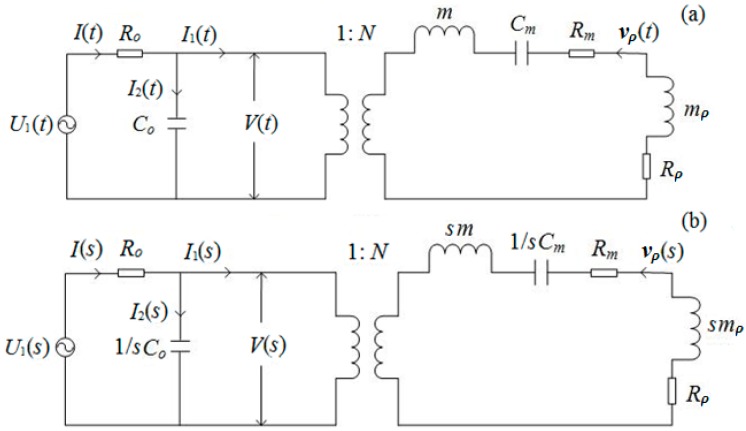
Equivalent circuits of a thin cylindrical transducer for electric–acoustic conversion: (**a**) equivalent circuit in the time domain and (**b**) equivalent circuit in the *s*-domain, where it is excited by a harmonic sinusoidal electric signal. $U_{1}(t)$ is the driving voltage source and *R*_o_ is its output resistance; *V*(*t*) is the voltage signal at the electric terminals of the source; $m_{\rho }$, $R_{\rho }$, $C_{m}$, *m*, $C_{o}$, *N*, and $R_{m}$ are the radiation mass, radiation resistance, elastic stiffness, mass, clamped capacitance, mechanical–electric conversion coefficient, and fraction force resistance of transducer, respectively; and $v_{\rho }(t)$ is the vibration speed at the transducer surface.

**Figure 3 micromachines-10-00804-f003:**
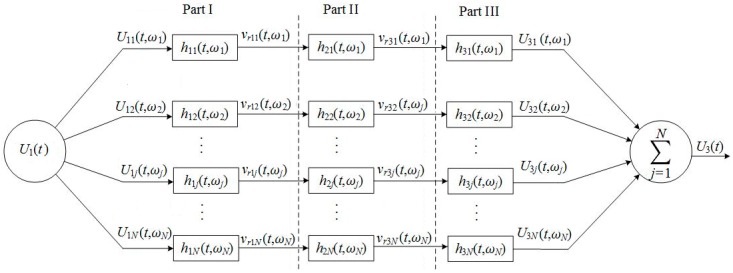
Schematic representation of a transmission network, which shows an acoustic-measurement process. *N* is the total number of components in the frequency spectrum of a driving electric signal. *U*_1*j*_ is the *j*th frequency component in the driving electric signal. *v_r_*_1*j*_ is the *j*th sinusoidsal frequency component of the vibration speed on the surface of the transducer. *v_r_*_3*j*_ is the *j*th sinusoidal frequency component in the acoustic signal arriving at the receiver transducer. *U*_3*j*_ is the *j*th frequency component of measured acoustic signal (i.e. electric signal at the electric terminals of the receiver transducers) created by the acoustic–electric conversion of the receiver transducer, where *j* = 1, 2, …, *N*.

**Figure 4 micromachines-10-00804-f004:**
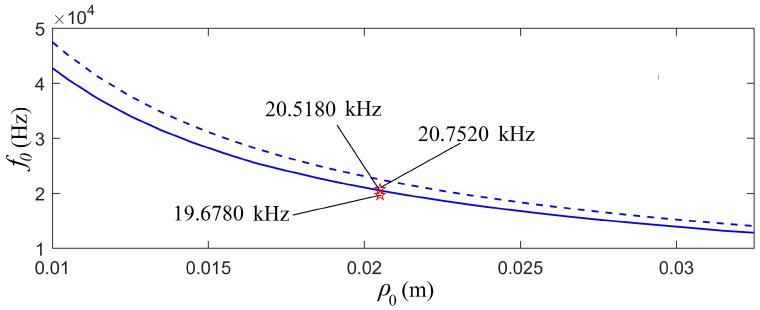
The relationship between the transducer’s center frequency versus its radius. The solid line is the center frequency for the case with a mechanical load (i.e. transducer was put in transformer oil) and the dashed line is the center frequency for the case of a free mechanical load (i.e. transducer was put in air or vacuum). The two star signs are the measured values for the case of the transducer with a mechanical load and $l_{t}$ = $\rho_{0}$/8.

**Figure 5 micromachines-10-00804-f005:**
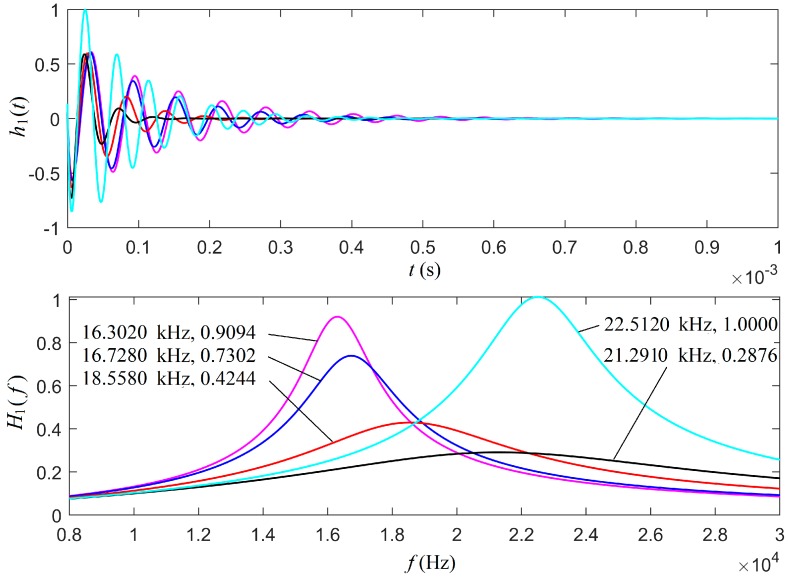
The impulse response and corresponding amplitude spectrum for a thin cylindrical transducer: (**a**) the impulse response and (**b**) the amplitude spectrum. The cyan line is the case of a free mechanical load. The other lines are for a mechanical load, where the magenta, blue, red, and black lines stand for the sinusoidal driving electric signals with frequency $f_{s}$ = 0.1$f_{0}$, 0.2$f_{0}$, 0.5$f_{0}$, and 1.5$f_{0}$, respectively, where $f_{0}$ = 20.5180 kHz. The first numerical value in Figure 5b is the center frequency and the second one is the maximal value of the amplitude spectrum.

**Figure 6 micromachines-10-00804-f006:**
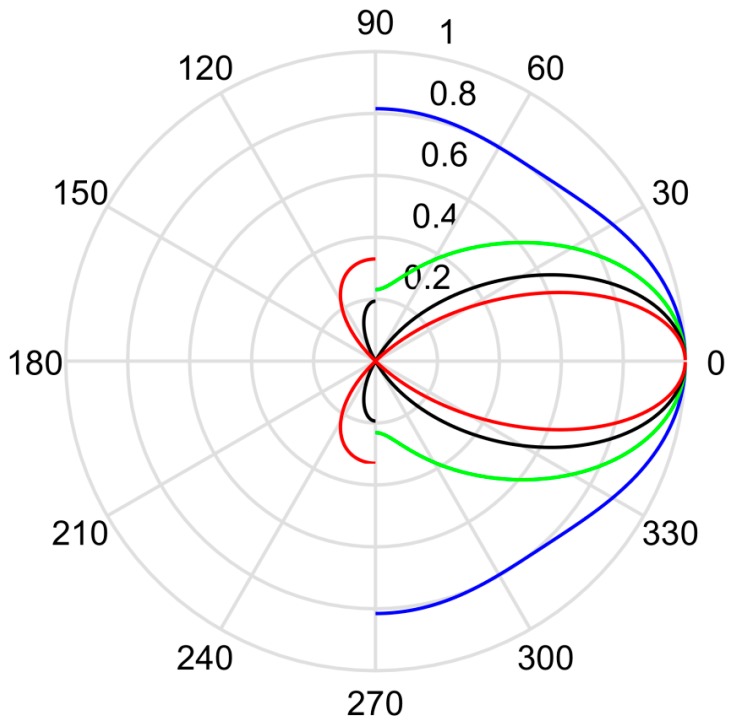
The directivity of thin cylindrical transducer with varied height of the cylindrical transducer, where $\rho_{0}$ = 20.50 mm. Red, black, green, and blue colors indicate the directivities of *H* = 40 mm, 60 mm, 80 mm, and 100 mm, respectively.

**Figure 7 micromachines-10-00804-f007:**
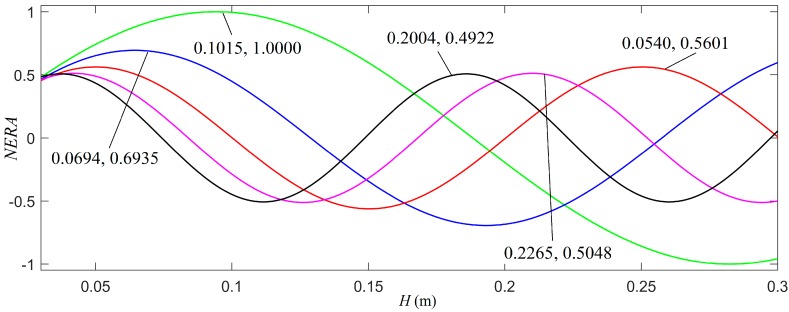
The relationship of the normalized effective radiating area (*NERA*) versus height (*H*) for several different directions, where $\rho_{0}$ = 20.5 mm. The green, blue, black, magenta, and red colors designate the cases of radiating direction *θ* = 20°, *θ* = 30°, *θ* = 40°, *θ* = 50°, and, *θ* = 60°, respectively. The first numerical value in Figure 8 is the height, *H*, of the transducer and the second one is the maximal value of the normalized effective radiating area. The value of *NERA* is the value of *ERA* normalized using the maximum of *ERA* at *θ* = 20°.

**Figure 8 micromachines-10-00804-f008:**
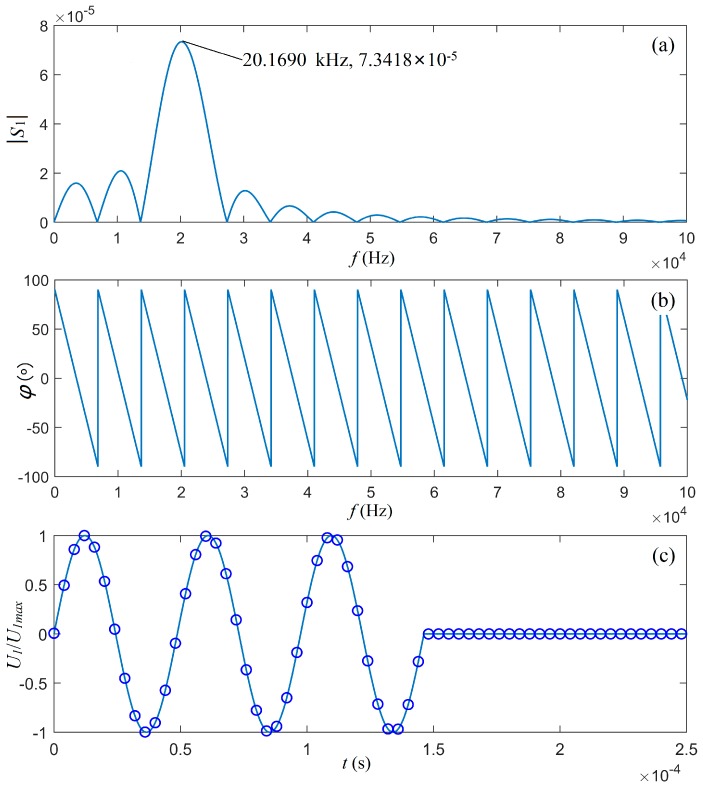
The waveforms of the gated sinusoidal driving electric signal with two cycles and $f_{s}$ = 20.5180 kHz, where $f_{s}=\omega_{s}/2\pi$. (**a**) The amplitude spectrum, (**b**) the phase spectrum, and (**c**) the waveform. The solid line in (c) was from the theoretical calculation and the cycle line was the synthesized waveform from discretized amplitude and phase spectra.

**Figure 9 micromachines-10-00804-f009:**
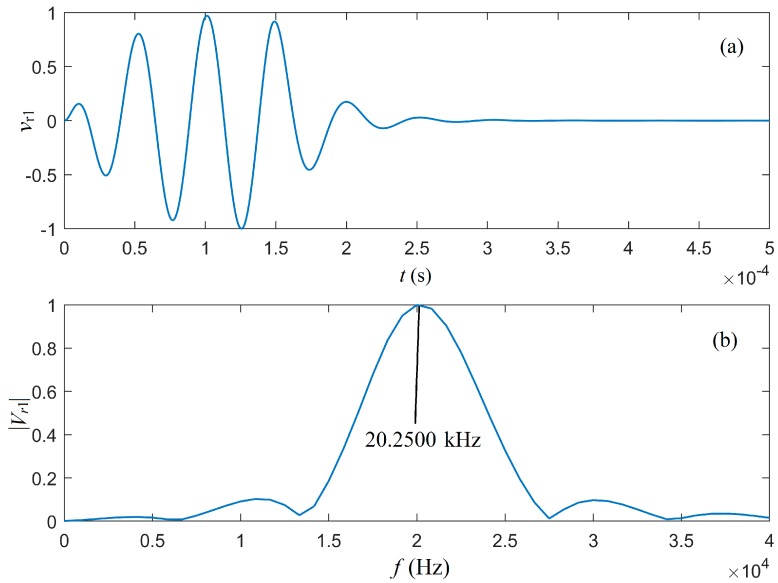
The waveform and amplitude spectrum of the acoustic signal radiated by the source transducer: (**a**) The normalized waveforms, which were the cumulative convolution of all frequency components in the network; and (**b**) the normalized amplitude spectrum of our transducer’s transient response model.

**Figure 10 micromachines-10-00804-f010:**
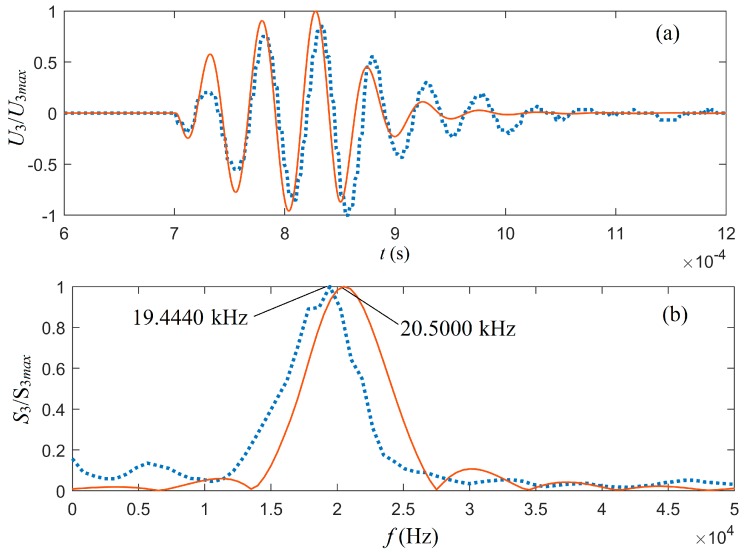
Normalized electric signals at the receiver transducer (part III of Figure 4): (**a**) the waveform and (**b**) the amplitude spectrum. $S_{3}$ stands for the amplitude spectrum corresponding to the electric-signals $U_{3}$ at the receiver transducer. Solid lines come from the theoretical calculation and dotted lines come from the experimental measurement.

**Figure 11 micromachines-10-00804-f011:**
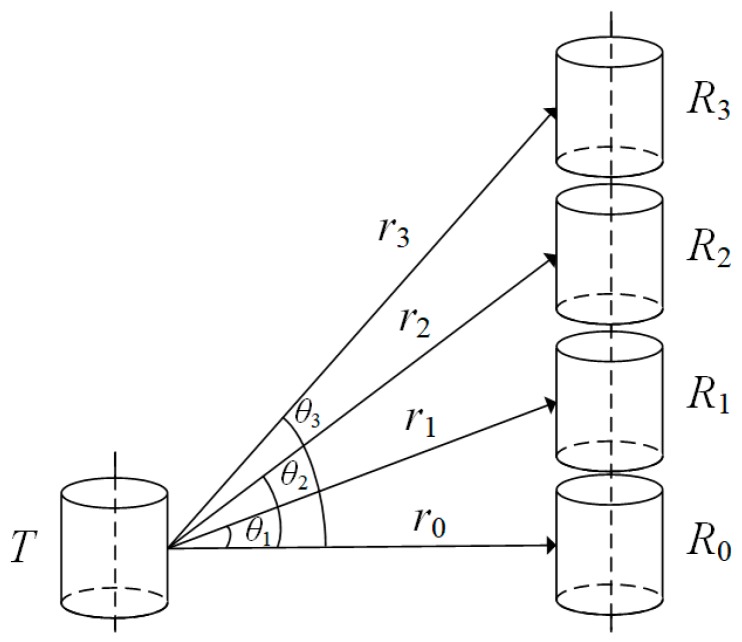
The acoustic-measurement system with a thin cylindrical transducer as a source and an array of four cylindrical transducers as the receivers, where *T* is the source transducer, *R_i_* is each receiver transducer in the transducers line array, *r_i_* is the distance from the source transducer *T* to the receiver transducer *R_i_* of the transducer line-array with {*i*} = {0, 1, 2, 3}, and *θ_j_* is the angle of *r_j_* with respect to *r*_0_ with {*j*} = {1, 2, 3}.

**Figure 12 micromachines-10-00804-f012:**
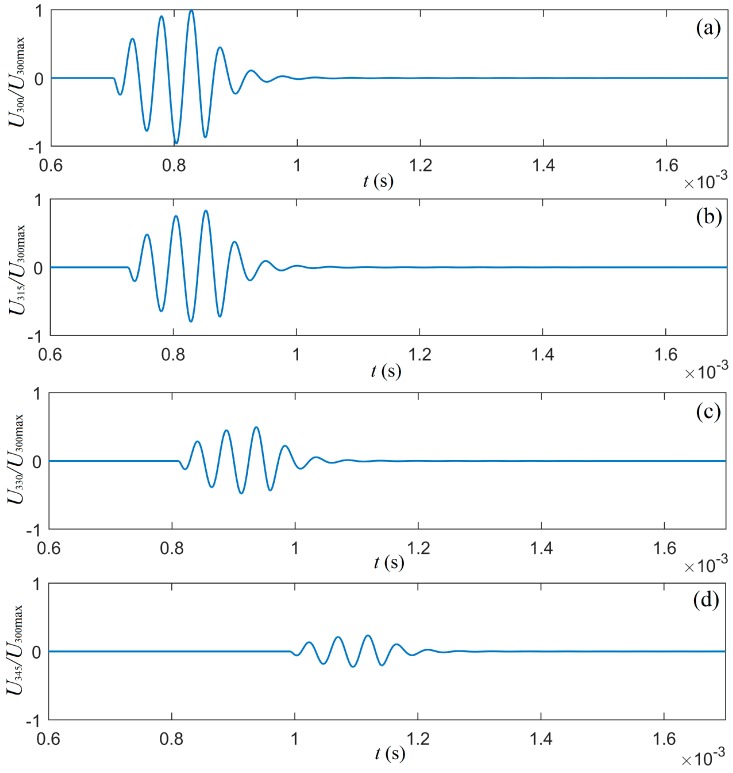
The signal waveforms at the electric-terminals of the receiver-transducer.

**Figure 13 micromachines-10-00804-f013:**
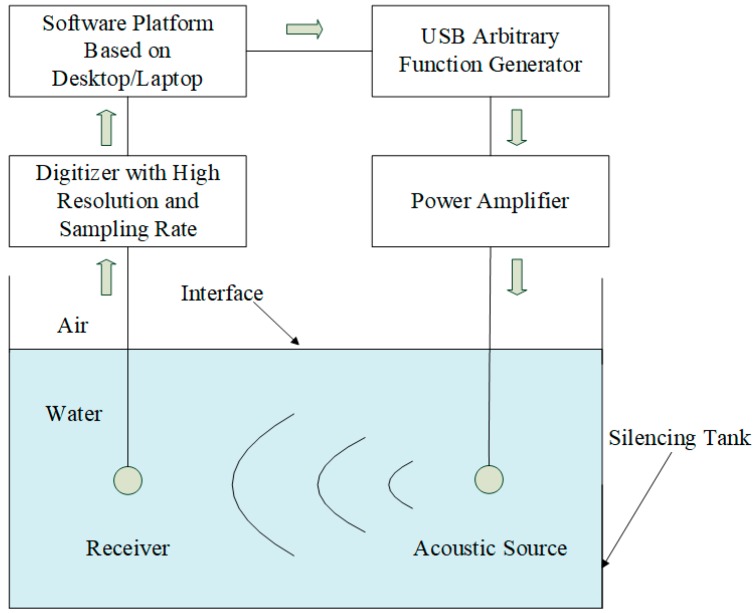
Schematic presentation of the experimental measurement system.

**Figure 14 micromachines-10-00804-f014:**
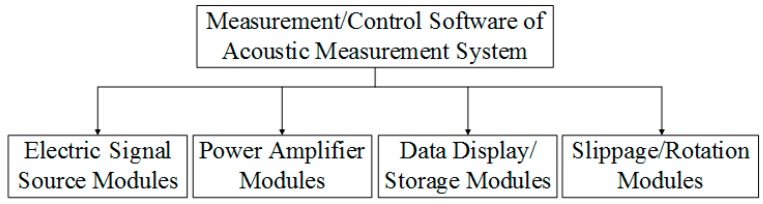
Structure flowchart of the modules for an electric signal source, power amplifier, date display/storage, and slippage/rotation.

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
