# Peer review of "Physical Characteristics of and Transient Response from Thin Cylindrical Piezoelectric Transducers Used in a Petroleum Logging Tool"

_micromachines, 2019, doi:10.3390/mi10120804_

Round 1

Reviewer 1 Report

The paper is general of interest and value and should be published after revision.

Comments:

[1] Scholarly papers on acoustic radiation from finite cylinders have been published by other authors, such as W. Williams, P.G. Bordoni, CWJS Lai, X.F. Wu and F.H. Fenlon;  yet these authors have not been referenced.

[2] the choice of the letter ρ for radius and mass density is awkward and should be changed.  There is also a ρm

[3] Ti is not defined in text in connection with Equ. 3.

[4] Both theoretical and experimental frequencies are given with 6 figure precision.  How is possible with short pulses?  What is the repeatability of such precision.  Moreover, what is the significance?

The paper presents the theoretical aspects in great detail but presents only some experimental data.  The title implies application to well logging bukt there is explanation or application given of how the results have been applied.

Reviewer 2 Report

This research paper explains about modeling, calculation, and experimental measurements, that have been achieved on enhanced understanding of the transient responses of the electric-acoustic and acoustic-electric conversions based on piezoelectric cylindrical transducers, widely used petroleum acoustic-logging.

-Abstract: The text must be carefully revised. Some sentences contain mistakes (in the abstract: very general statements) whereas some sentences must be reworded as the English is “meaningless”. I strongly recommend that the authors retain the services of a professional editor. There are many reputable companies that offer these services.

Section 1- Introduction is poorly written. Proper references need to be used rather than using others. For example, too many references are loaded in Line no. 38 [5-12] and Line no. 61 [31-42]. This need to be taken care. Language can be improved. The sentences are half constructed or incomplete in a way that the readers are expected to fend for themselves in order to understand their meaning.

Section 2- Theory and Modeling: No references are provided. The theory is well known and it found in many classical books. There are too many equations which are repeated from literature. If it is not derived by authors, it is better to curtail few equations.

Overall the paper appears to be of satisfactory quality. Only if the language was better, it would require lesser effort to understand and comprehend what the writer intends to convey. I would recommend this paper to be published after minor revisions.

Round 2

Reviewer 1 Report

ready for publication

This manuscript is a resubmission of an earlier submission. The following is a list of the peer review reports and author responses from that submission.